



**No nitrogen fixation in the Bay of Bengal?**
Carolin R. Löscher[1,2], Wiebke Mohr[3], Hermann W. Bange[4], Donald E. Canfield[1]
[1]Nordcee, Department of Biology, University of Southern Denmark, Odense, Denmark
[2]D-IAS, University of Southern Denmark, Odense, Denmark
[3]Max Planck Institute for Marine Microbiology, Bremen, Germany
[4] GEOMAR Helmholtz Center for Ocean Research Kiel, Kiel, Germany
Correspondence to cloescher@biology.sdu.dk
The Bay of Bengal (BoB) has long stood as a biogeochemical enigma with subsurface waters containing
extremely low, but persistent, concentrations of oxygen in the nanomolar range which -for some, yet
unconstrained reason- are prevented from becoming anoxic. One reason for this may be the low productivity
of the BoB waters due to nutrient limitation, and the resulting lack of respiration of organic material at
intermediate waters. Thus, the parameters determining primary production are key to understanding what
prevents the BoB from developing anoxia. Primary productivity in the sunlit surface layers of tropical oceans
is mostly limited by the supply of reactive nitrogen through upwelling, riverine flux, atmospheric deposition,
and biological dinitrogen ($N_2$) fixation. In the BoB, a stable stratification limits nutrient supply via upwelling
in the open waters, and riverine or atmospheric fluxes have been shown to support only less than one quarter
of the nitrogen for primary production. This leaves a large uncertainty for most of the BoB's nitrogen input,
suggesting a potential role of $N_2$ fixation in those waters.
Here, we present a survey of $N_2$ fixation and carbon fixation in the BoB during the winter monsoon season.
We detected a community of $N_2$ fixers comparable to other OMZ regions, with only a few cyanobacterial
clades and a broad diversity of non-phototrophic $N_2$ fixers present throughout the water column (samples
collected between 10 m and 560 m water depth). While similar communities of $N_2$ fixers were shown to
actively fix $N_2$ in other OMZs, $N_2$ fixation rates were below the detection limit in our samples covering the
water column between the deep chlorophyll maximum and the OMZ. Consistent with this, no $N_2$ fixation
signal was visible in $\delta^{15}N$ signatures. We suggest that the absence of $N_2$ fixation may be a consequence of a
micronutrient limitation or of an $O_2$ sensitivity of the OMZ diazotrophs in the BoB. To explore how the onset
of $N_2$ fixation by cyanobacteria compared to non-phototrophic $N_2$ fixers would impact on OMZ $O_2$
concentrations, a simple model exercise was carried out. We observed that both, photic zone-based and
OMZ-based $N_2$ fixation are very sensitive to even minimal changes in water column stratification, with





stronger mixing increasing organic matter production and export, which would exhaust remaining $O_2$ traces
in the BoB.
**Introduction**
Primary production in large areas of the surface ocean is limited by the availability of fixed nitrogen (Moore
et al., 2013). This deficiency in nitrogen creates a niche for dinitrogen ($N_2$) fixation, an energy-costly process
carried out only by certain prokaryotes, also referred to as diazotrophs, which are phylogenetically highly
diverse. $N_2$ fixation in the ocean has been described quantitatively as most important in the oligotrophic
surface waters of the subtropical gyres (Sohm et al., 2011;Luo et al., 2012;Wang et al., 2019) where
cyanobacterial $N_2$ fixers dominate. Over the last decade, the development of novel molecular tools revealed
that non-cyanobacterial $N_2$ fixers are widely distributed throughout ocean waters (Farnelid et al.,
2011;Farnelid et al., 2013;Fernandez et al., 2011;Luo et al., 2012;Riemann et al., 2010;Zehr et al., 1998) and
sediments (Fulweiler et al., 2007;Andersson et al., 2014;Bertics et al., 2013;Gier et al., 2017;Gier et al., 2016).
Their quantitative importance for global $N_2$ fixation, however, is not yet clear. In oxygen minimum zones
(OMZs) of the eastern tropical North and South Pacific Ocean, hypoxic basins in the Californian Bay and the
Arabian Sea, those $N_2$ fixers form a unique community consisting of different clades of proteobacteria,
clostridia, spirochaetes, chlorobia, and methanogenic archaea (Christiansen and Loescher,
2019;Dekaezemacker et al., 2013;Fernandez et al., 2011;Gaby et al., 2018;Gier et al., 2017;Goebel et al.,
2010;Halm et al., 2012;Hamersley et al., 2011;Jayakumar et al., 2012;Jayakumar et al., 2017;Löscher et al.,
2014). In contrast, cyanobacterial $N_2$ fixers and diatom-diazotroph-associations (DDAs), which are commonly
considered the most important $N_2$ fixers in the surface ocean, were either absent or were detected only in
low abundances in OMZs (Turk-Kubo et al., 2014;White et al., 2013;Jayakumar et al., 2012). Both, the
presence of diazotrophs clustering with proteobacteria, clostridia, spirochaetes, chlorobia, and
methanogenic archaea, and the underrepresentation of cyanobacterial and DDA $N_2$ fixers could thus be
considered characteristic for OMZ environments.
Nutrient stoichiometry and model predictions (Deutsch et al., 2007) suggest that oxygen-deficient waters are
a potentially important niche for $N_2$ fixation. Based on this suggestion, several studies have focused on $N_2$
fixation in the large and persistent OMZ of the eastern tropical South Pacific. In this region $N_2$ fixation rates
vary, however, with maximum rates of 840 µmol N $m^{-2}d^{-1}$ detected in nitrogenous sulfidic waters off the
coast of Peru (Löscher et al., 2014), and 117 µmol N $m^{-2}d^{-1}$ in the oxygen-depleted zone (Bonnet et al., 2013).
Low $N_2$ fixation rates close to the detection limit were reported from the same area (Chang et al., 2019),
another set of $N_2$ fixation rates estimated from sediment trap analyses were in the range of 0-23 µmol N
$m^{-2}d^{-1}$ (Knapp et al., 2016). Taken together, these rates suggest either a strong temporal variation or spatial





patchiness. A similar variation in $N_2$ fixation rates was described for the eastern tropical North Pacific ranging
from close to the detection limit in the OMZ (Jayakumar et al., 2017) up to 795 µmol N $m^{-2}d^{-1}$ in episodic
diazotroph blooms (White et al., 2013)). This apparent temporal or spatial variation in $N_2$ fixation rates may
originate from unresolved environmental controls on $N_2$ fixation and makes it difficult to quantify $N_2$ fixation
in OMZ waters.
The Bay of Bengal (BoB) is a region with a strong seasonality driven by the Asian monsoon system. Massive
rainfall and river discharge with maximum freshwater inputs in September (e.g. Mahadevan (2016)) cause a
considerable lowering of surface water salinity during and after the monsoons throughout the whole basin
(Subramanian, 1993). This, together with increased surface water temperatures, leads to intensive and
persistent stratification of the water column (Kumar et al., 2004), restricting nutrient fluxes to the surface
from below and promoting a strong OMZ (McCreary et al., 2013;Paulmier and Ruiz-Pinto, 2009;Sarma, 2002)
with minimum oxygen ($O_2$) concentrations in the lower nanomolar range (Bristow et al., 2017).
The potential importance of $N_2$ fixation in the BoB can be derived from a simple N budget estimate with an
overall N loss of 7.9 ± 0.6 Tg N $yr^{-1}$ and N sources other than $N_2$ fixation of 3.15 ± 2,25 Tg N $yr^{-1}$ (Tab. 1, data
from (Naqvi, 2008;Naqvi et al., 2010;Bristow et al., 2017;Singh et al., 2012;Krishna et al., 2016;Srinivas and
Sarin, 2013;Suntharalingam et al., 2019)). This implies a deficit of 4.7 ± 2.4 Tg N $yr^{-1}$ within the given range of
uncertainty indicating the potential importance of $N_2$ fixation assuming a coupling of nitrogen loss and $N_2$
fixation as proposed by (Deutsch et al., 2007). Naqvi et al. (2010) proposed $N_2$ fixation to contribute 1 Tg N
$yr^{-1}$ in the BoB, while Srinivas and Sarin (2013) interpolated a contribution of 0.6- 4 Tg N $yr^{-1}$ from phosphate
availability. Measurements of $N_2$ fixation rates from the BoB are not available, isotope analysis of sediment
trap samples indeed suggests that the BoB is a site of active $N_2$ fixation. Indeed, the composition of the
organic material produced in BoB surface waters is characterized by a high portion of biogenic opal (20%)
and a low $\delta^{15}N$ nitrate signal (3.2 - 5‰, Gaye-Haake et al. (2005)). This points towards a production of a
considerable part of organic matter produced by diatoms in symbiotic association with or in close proximity
to diazotrophs (Subramaniam et al., 2008). Only few studies report the presence of diazotrophs including
*Trichodesmium* in the BoB (Wu et al., 2019;Shetye et al., 2013;Sahu et al., 2017;Jyothibabu et al.,
2006;Mulholland and Capone, 2009), with only one of them using a functional gene approach.
To investigate the diazotrophic community and to quantify $N_2$ and carbon fixation in the BoB OMZ, we used
a combination of gene sequencing and quantification, rate measurements, isotope tracing and box modeling.



## Methods


Geochemical sampling
Samples were collected from the top 500 m of the water column during a cruise with the ORV *Sagar Kanya*
to the BoB during the winter monsoon in January 2014. Seawater samples were collected using 5 L and 30 L-
Niskin bottles on a CTD- rosette equipped with a Seabird SBE 43 oxygen sensor and a WET Labs ECO-AFL/FL
chlorophyll sensor as previously described in Bristow et al. (2017). To resolve oxygen dynamics below the
Seabird sensor's detection limit a STOX (Switchable Trace amount OXygen) amperometric oxygen sensor was
used (Revsbech et al., 2009), which had a detection limit of $7 - 12$ nmol $L^{-1}$ during this sampling campaign
(Bristow et al., 2017). Nutrients, including nitrate, nitrite and phosphate were determined according to
Grasshoff (1999).

$N_2$/C-fixation rate measurements
Seawater was collected from the Niskin bottles and filled into 2.4-L glass bottles or 2.8-L polycarbonate
bottles for (near-) anoxic and all other (oxic) waters, respectively. Bottles were capped with black rubber
stoppers (anoxic waters) or Teflon-coated butyl rubber septa (oxic waters). Incubations were performed with
the method developed by Mohr et al. (2010) as described in (Grosskopf et al., 2012). Batches of $^{15}N_2$ gas
(Cambridge Isotopes, USA) –enriched water was prepared with degassed water from two – three of the six
sampling depths. Each incubation bottle was supplemented with 50 mL of the $^{15}N_2$-enriched seawater.
Discrete samples for the measurement of the $^{15}N_2$ concentration were taken from each incubation bottle and
were measured by membrane-inlet mass spectrometry (MIMS). Final $^{15}N_2$ enrichments were on average 1.65
atom % $^{15}N$. For carbon fixation measurements, $NaH^{13}CO_3$ was dissolved in sterile MilliQ water (1g/117mL),
and 5 mL were added to each incubation (~8 atom% final, based on total DIC of 2.2 mM). Bottles with water
from the upper two depths were kept in surface seawater-cooled on-deck incubators. Bottles from the lower
depths were incubated at 13-15°C in the dark. Incubations were stopped after approximately 24 h (samples
with less than 20h incubation time were excluded from our analysis). Volumes between 2.1 and 2.7 L of
seawater were filtered onto pre-combusted (450°C, 4-6 hours) 25 mm diameter GF/F filters (Whatman, GE
Healthcare, Chalfont St Gile, UK) under gentle vacuum (200 mbar). Filters were either frozen at -20°C and
oven-dried prior to processing or oven-dried (50°C) directly for 24 h and stored dry until analysis. Untreated
seawater was filtered and prepared as described above to obtain background natural abundance values. For
elemental and isotopic analysis, GF/F filters were acidified over fuming HCl overnight in a desiccator to
remove inorganic C. Filters were then oven-dried for 2 hours at 50°C and pelletized in tin cups. Samples for
particulate organic carbon and nitrogen (POC and PON) and C and N isotopic composition were analyzed on



an Elemental Analyzer Flash EA 1112 series (Thermo Fisher) coupled to a continuous-flow isotope ratio mass
spectrometer (Finnigan Delta Plus XP, Thermo Fisher). A table of $N_2$ and C fixation rate measurements is given
in the supplementary material. Data sets were deposited on PANGAEA.

Molecular methods
Nucleic acid samples were collected at stations 1, 4 and 5 (Fig. 1). Between 5 and 27 L of seawater were
filtered in two size fractions (3 μm and 0.22 μm pore size, Supor PES membrane disc filters; Pall, Portsmouth,
UK), exact filtration volumes were recorded. Filters were stored in 2.7 mL sucrose lysis buffer at −20 °C.
DNA was extracted using an established protocol based on a phenol/chloroform extraction (Giovannoni et
al., 1996). The quality and concentration of the purified DNA was checked spectrophotometrically and using
the Quant-iT PicoGreen dsDNA kit (Invitrogen, Carlsbad, USA).
A metagenome from the deep chlorophyll maximum (DCM, 84m water depth) at station 4 was Illumina HiSeq-
sequenced using a 2x125bp read length on a NexteraXT library at the Institute for Clinical Microbiology
(IKMB) at Kiel University, Germany. Sequencing resulted in 321Mbp. Sequences were analyzed using the
MetPathways pipeline (Konwar et al., 2013), a modular annotation and analysis pipeline for predicting
diversity and metabolic interaction from environmental sequences consisting of a quality control, an open
reading frame prediction and annotation, diversity analysis, and environmental pathway reconstruction.
Phylogenetic identification of OTUs was derived via a comparison with the RefSeq and Greengenes databases
(DeSantis et al., 2006). After quality check, 6,454 sequences of ribosomal RNA were identified, 622,286
sequences (27.56%) of proteins with known functions, and 1,628,841 sequences (72.15%) were predicted
proteins with unknown function.
*nifH* gene amplification was performed using a nested PCR protocol (Zehr et al., 1998). PCRs were performed
using the GoTaq kit (Promega, Fitchburg, USA) adding one additional μL BSA (20 mg mL$^{-1}$ (Fermentas,
Waltham, USA). The TopoTA Cloning® Kit (Invitrogen, Carlsbad, USA) was used for cloning of PCR amplicons,
according to the manufacturer's protocol. Sanger sequencing (340 *nifH* sequences) was performed by the
Institute of Clinical Molecular Biology, Kiel, Germany. Negative controls were performed using the PCR
mixture as described without template DNA; no amplification was detected. Samples from the particulate
fraction >3 μm were consistently negative for *nifH* gene copies and were thus not further investigated.
Sequences were ClustalW aligned in MEGA 7 (Kumar et al., 2016) and a maximum likelihood tree was
constructed on a 321 base pair fragment. Reference sequences were obtained using BlastX on the NCBI



database. Sequences were submitted to Genbank, submission ID 2245434. The metagenome has been
submitted to the NCBI's sequence read archive, accession number SRR9696254.
Quantitative real time PCRs for *nifH* were performed using cluster specific TaqMan-probe qPCRS as described
in Löscher et al (2014), with primers, probes, environmental standards and PCR conditions as presented in
table S1. Samples were run in duplicates on a Biorad qPCR machine (Biorad, Hercules, USA).

Box model exercise
We used a simplistic five-box representation of an upwelling system with a deep and intermediate water iron
source, with primary and export production as well as respiration derived from the original models (Canfield,
2006;Boyle et al., 2013). The model was used to distinguish a $N_2$ fixation state of the BoB and a non-$N_2$ fixation
state with primary production driven by recycled dissolved nitrogen compounds. In contrast to the previous
model versions, we applied a non- Redfield-based $N_2$ fixation scenario. Ammonia concentrations were set to
zero in all boxes, in accordance with our direct measurements. Fe concentrations were set to 0.1 µmol $L^{-1}$ in
the deep and intermediate water boxes and 0.00044 µmol $L^{-1}$ in the productive zone (Grand et al.,
2015a;Grand et al., 2015b). Oxygen concentrations were adjusted to our measurements with 220, 0.02 and
50 µmol $L^{-1}$ in the surface, OMZ  and deep water, respectively (Bristow et al., 2017). Phosphate and nitrate
concentrations were taken from our measurements with phosphate concentrations of 0, 2.7 and 2.5 µmol $L^{-1}$
$^1$ in the surface, OMZ and deep boxes, respectively; and oxidized nitrogen compounds (nitrate+ nitrite) at a
concentration of 0, 38 and 35 µmol $L^{-1}$ in the surface, intermediate and deep boxes, respectively. Further
information on the model stoichiometry is given in the supplementary material.

**Results and discussion**
We explored the diversity, distribution and activity of $N_2$ fixing microbes and carbon fixers in the OMZ of the
northern BoB during the Northeast monsoon in January 2014. During the time of the cruise, low sea surface
temperatures (SST) and low surface water salinity reaching from the coasts of India, Bangladesh and
Myanmar southwards to approximately 16°N were present (Figure 1A, B). At the coast, this low salinity/low
SST plume co-occurred with increased chlorophyll concentrations (Fig 1C), thus suggesting a stimulation of
primary production by waters possibly of riverine origin (Fig 1C). This is in line with earlier suggestions of
riverine nutrient runoff promoting primary production close to the shelf, where nutrients are consumed
rapidly thus preventing their offshore transport (Kumar et al., 2004;Singh et al., 2012;Singh and Ramesh,
2011;Krishna et al., 2016). Chlorophyll concentrations in the BoB during the time of the cruise detected via
satellite monitoring ranged between 0.08 mg $m^{-3}$ in open waters and 15 mg $m^{-3}$ at the northern coast and



were consistent with previous in-situ measurements during low productivity periods in the BoB (Prasanna
Kumar et al., 2010).
The sampling stations were located offshore in the central BoB (Fig. 1), where waters were strongly stratified
with low sea surface salinity, but warmer SST compared to the coast, and a steep oxycline reaching $O_2$
concentrations close to anoxia at around 100 m water depth. No in-situ chlorophyll measurements are
available from the cruise, but a fluorescence sensor attached to the CTD showed a maximum of up to 0.8 mg
$m^{-3}$ between 32-90 m water depth (Fig. 2). Satellite derived chlorophyll concentrations in the coastal BoB
were in the range from 0.08 to 0.35 mg $m^{-3}$, slightly higher than in a previous study of this region (0.06 mg $m^{-3}$
$^3$, Kumar et al. (2002)). Carbon fixation rates ranged between 286-1855 nmol C $L^{-1}$ $d^{-1}$ at the depth of the DCM
(Fig. 2), however, our rate measurements did not cover the water column above 60 m water depth where
rates may have been higher. Consistent with previous descriptions of primary producers at our study site
(Loisel et al., 2013) and with satellite imaging (Fig. S1), we identified cyanobacteria related t*o Synechococcus*
and *Prochlorococcus* as the most abundant primary producers in the in our metagenome from the BoB DCM,
accounting for 3.3% of OTUs while eukaryotic phytoplankton accounted for only 0.3% of OTUs (Table S2).
Comparable to chlorophyll, particulate organic carbon (POC, Tab. S3; see also Fig. S2 for a distribution of POC
in the BoB) concentrations were low, ranging between 4.96 and 7.84 µmol C $L^{-1}$ in surface waters, and
resulting in an average POC:chlorophyll ratio of 68:1 to 115:1 at the depth of the DCM (Fig. 1). This ratio, is
comparable to POC:chlorophyll ratios reported from cyanobacteria-dominated communities (74:1–126:1;
e.g., (Lorenzoni et al., 2015;Sathyendranath et al., 2009)), but it is higher compared to other OMZ regions
(e.g. 50:1 in the eastern tropical South Pacific (Chavez and Messié, 2009;Chavez et al., 1996)). Similarly,
carbon fixation rates were 1-2 orders of magnitude lower compared to the Arabian Sea, the tropical South
Pacific and tropical Atlantic (e.g. Longhurst et al. (1995)). While our POC concentrations from DCM are one
order of magnitude higher than the satellite-derived POC estimates (Fig. S2) from surface waters indicating
that primary production in surface waters was not higher than in the DCM, it must be noted that our
measurements did not cover the entire mixed layer and are thus likely a rather conservative minimum
estimate.
$N_2$ fixation in the upper water column and the oxycline
Based on the dissolved inorganic nitrogen ($NO_3^-$ + $NO_2^-$) to phosphate ($PO_4^{3-}$) ratio which has a negative
intercept with the y-axis (Fig. 3; Benitez-Nelson (2000)), the BoB waters were nitrogen limited during the
cruise. This nitrogen limitation would be expected to create a niche for $N_2$ fixation, but except for two samples



for which in both cases only one out of three technical replicates showed an isotope enrichment, $N_2$ fixation
rates were below the detection limit (Tab. S1). Consistent with this, $\delta^{15}N$ signatures of both the nitrate and
the particulate organic nitrogen (PON) pool were only slightly decreased in the top 100 m of the water column
(Fig. S3), thus not suppporting active $N_2$ fixation. Several clusters of $N_2$ fixing microbes were, however,
identified by screening for the key functional marker gene *nifH* (Fig. 4). Only a few *nifH* sequences were
associated with cyanobacteria commonly abundant in ocean surface waters. This pattern seems to be typical
for OMZ areas (Fernandez et al., 2011;Jayakumar et al., 2012;Löscher et al., 2014) and for the eastern Indian
Ocean (Wu et al., 2019), where cyanobacterial *nifH* sequences are also rare. Similar to earlier studies, which
identified *Trichodesmium* in BoB surface waters (Bhaskar et al., 2007;Hegde, 2010;Wu et al., 2019), we
detected *nifH* copies related to *Trichodesmium* in our samples, both by sequencing and by qPCR (Fig. 4, Tab.
S4). These sequences clustered closely to *Trichodesmium-nifH* previously recovered from the Arabian Sea
(Jayakumar et al., 2012;Mazard et al., 2004), where those $N_2$ fixers were found in low abundances, but
possibly actively fixing $N_2$ as indicated by *nifH* presence in a cDNA library. No sequences related to the
different groups of unicellular cyanobacterial diazotrophs (UCYN-A, -B, or –C; Zehr et al. (2001)) were present
in our *nifH* dataset. UCYN-A and UCYN-B have previously been found in the Arabian Sea, but only at
oligotrophic stations with warm water temperatures >30°C (Mazard et al., 2004). While UCYN-A may occur
at temperatures below 25°C, *Trichodesmium* and UCYN-B may be limited by the water temperatures at our
sampling stations, which were possibly too low with around 25°C. *Trichodesmium* is usually abundant in high-
iron input regions such as the tropical Atlantic Ocean (Martínez-Pérez et al., 2016). The absence of
*Trichodesmium* and other cyanobacterial $N_2$ fixers may thus also result from an insufficient iron source
(Moore et al., 2013). Additionally, light limitation due to severe atmospheric pollution (known as the 'South
Asian Brown Cloud') which lasts over the BoB from November to May (e.g. Ramanathan et al. (2007)) may
influence the distribution of cyanobacteria in the BoB (Kumar et al., 2010). While earlier studies also detected
*Chaetoceros* (Bhaskar et al., 2007;Hegde, 2010;Wu et al., 2019), a diatom known to live in association with
diazotrophs, no diatom-associated $N_2$ fixers could be identified from our sequences. Thus our data does not
directly support previous suggestions of those specific diazotrophs producing low $\delta^{15}N$ nitrate signatures
along with high opal concentrations previously detected in sediment trap samples (Gaye-Haake et al., 2005).

$N_2$ fixation in the OMZ
In the OMZ, we detected again the genetic potential for $N_2$ fixation, but $N_2$ fixation rates were below the
detection limit and $\delta^{15}N$ signatures of nitrate and PON indicated nitrogen loss instead of $N_2$ fixation (Fig. S3).



The community of N$_2$ fixers in the BoB consisted mostly of the non-phototrophic, proteobacterial
representatives of *nifH-* clusters I and III (Fig. 4), most of them related to previously identified OMZ
diazotrophs (Fernandez et al., 2011;Jayakumar et al., 2012;Löscher et al., 2014).
A statistical comparison of BoB *nifH* sequences with OMZ diazotroph communities from the Arabian Sea, the
ETSP, ETNP and hypoxic basins in California Bay revealed a strong similarity suggesting that certain
diazotrophs are characteristic for OMZs (Fig. 5). Those typical OMZ-clusters include uncultured ɣ-, ꝺ- and ɛ-
proteobacteria and clostridia. Only one cluster was uniquely represented in the BoB and absent from the
other OMZ datasets, with only three individual sequences related to *Azotobacter chroococum*. Another
difference between the BoB and in the other OMZ diazotroph communities was the composition of Cluster
IV *nifH* sequences, which are present but cluster in different groups as compared to for instance the Arabian
Sea Cluster IV community. It is, however, unlikely that Cluster IV diazotrophs are important for N$_2$ fixation in
the BoB or other OMZs because they were never shown to be transcribed (Fernandez et al., 2011;Jayakumar
et al., 2012;Löscher et al., 2014) and Cluster IV-*nif* is generally considered to encode non-functional *nif* or
paralogous sequences (Gaby and Buckley, 2014;Angel et al., 2018). In addition, the presence of Cluster IV
*nifH* sequences has previously been ascribed to PCR-contamination (Zehr et al., 2003). Thus, the importance
of this cluster for N$_2$ fixation in OMZs is generally debatable and the different composition of the Cluster IV
diazotroph community does likely not explain the absence of N$_2$ fixation in the BoB.
While diazotroph communities highly similar to the identified BoB diazotrophs promote active N$_2$ fixation in
other OMZ waters, we have no consistent indication for N$_2$ fixation in the BoB (Table S1). One explanation
for the absence of N$_2$ fixation could be the sensitivity of the BoB OMZ diazotrophs to O$_2$ as opposed to the
relative O$_2$ tolerance of cyanobacterial N$_2$ fixers. We identified BoB diazotrophs closely related to cultivated
N$_2$ fixers, including *Vibrio diazotrophicus* and *Desulfonema limnicola*, which fix N$_2$ only under strictly
anaerobic conditions (Urdaci et al., 1988;Bertics et al., 2013;Gier et al., 2016). Further, communities of
diazotrophs from other OMZs highly similar to the BoB diazotroph community were described to transcribe
their *nifH* gene and to actively fix N$_2$ only under strictly anoxic or anoxic-sulfidic conditions (Löscher et al.,
2016;Löscher et al., 2014;Jayakumar et al., 2012;Jayakumar et al., 2017), and are unable to fix N$_2$ in the
presence of even minimal concentrations of O$_2$ (reviewed in Bombar et al. (2016)). N$_2$ fixation in our samples
(Tab. S1) may therefore be directly inhibited by the detected traces of O$_2$. Thus, our data suggests that even
only nanomolar O$_2$ concentrations such as present in the BoB may prevent non-phototrophic N$_2$ fixers from
actively fixing N$_2$, which could ultimately limit the supply of new nitrogen to the BoB.
Role of Fe and mesoscale activities (eddies)



The high iron (Fe) requirement of $N_2$ fixing microbes (60 times higher compared to other marine organisms,
Gruber and Galloway (2008)) limits $N_2$ fixation in large parts of the ocean (Moore et al., 2013). However,
aeolian Fe fluxes to surface waters of the southern BoB were estimated to be comparable to those detected
underneath Saharan dust plumes in the Atlantic (290 ± 70 µmol m$^{-2}$ yr$^{-1}$; Grand et al. (2015a)). Indeed,
dissolved Fe (dFe) accumulates in the BoB OMZ reaching comparably high concentrations of up to 1.5 nM
(Grand et al., 2015b;Chinni et al., 2019). In surface waters, dFe concentrations were described to range from
0.4 nM in the area of the cruise to up to 0.5 nM towards the north of the BoB, with increasing concentrations
coinciding with decreasing salinity north of 15°N (Grand et al., 2015a;Grand et al., 2015b;Chinni et al., 2019).
While the reported Fe concentrations do not indicate Fe limitation of $N_2$ fixation in the OMZ, surface primary
production and $N_2$ fixation may be limited by any other micro-nutrient. Indication for such a limitation can
be derived from eddy-induced Ekman pumping, mesoscale dynamics and the summer monsoon current have
been shown to trigger plankton blooms with high productivity (Jyothibabu et al., 2015;Vinayachandran and
Mathew, 2003;Chen et al., 2013;Fernandes et al., 2009) possibly induced by upwelling of certain nutrients to
surface waters. Besides locally increasing surface water chlorophyll concentrations, erosion of the strong
stratification and subsequent nutrient input to surface waters result in a change of phytoplankton size class
(Prasanna Kumar et al., 2004). While usually smaller phytoplankton dominate the primary producer pool (60
– 95 % of the total chlorophyll), the contribution of larger phytoplankton has been observed to double in the
regions influenced by the summer monsoon current and in mesoscale eddies, which impacts the vertical
organic carbon flux in the BoB temporally and locally (Jyothibabu et al., 2015;Prasanna Kumar et al.,
2004;Huete-Ortega et al., 2010;Gomes et al., 2016). The resulting increase of organic matter production, the
modified composition of organic matter (i.e. production fresh and labile POM), a faster export and
subsequent respiration could promote anoxic OMZ conditions in the BoB. This may subsequently allow for
$O_2$-sensitive processes to take place, which may include $N_2$ fixation and nitrogen loss processes (Johnson et
al., 2019), locally or regionally. Rapid changes in dissolved $O_2$ induced by increased surface productivity and
organic matter export were reported in the context of mesoscale water mass dynamics in the BoB (Johnson
et al., 2019), and also in other eddy systems in the Atlantic, which showed rapid $O_2$ exhaustion in otherwise
oxic waters (Fiedler et al., 2016;Löscher et al., 2015). Episodes of increased biological productivity have also
been reported from the BoB during both the pre-southwest monsoon and northeast monsoon (Kumar et al.,
2004). Under those scenarios, large parts of the BoB's surface waters exhibited a strong $p$CO$_2$ undersaturation
compared to the atmosphere (~350 µatm), resulting in an air-sea $p$CO$_2$ gradient sometimes exceeding 100
µatm. This gradient is explainable only by an increase in biological primary production fueled by temporal
external nutrient input (Kumar et al., 2004). As Singh et al. (2012) pointed out, these high productivity



episodes cannot be explained by riverine or atmospheric deposition of nutrients alone, but that upwelling or
$N_2$ fixation would be required to sustain the nitrogen demand.

Feedbacks between $N_2$ fixation and OMZ intensity
We used a simple model to test the conditions allowing for $N_2$ fixation in the surface waters and in the OMZ
of the BoB, and the interplay of $N_2$ fixation with primary production in response to changes in stratification
(i.e. upwelling). We further explored in how far $N_2$ fixation controls $O_2$ concentrations in the BoB OMZ. We
simulated a nitrate-driven primary production, and a $N_2$ fixation-dependent primary production, which is
representative of  $N_2$ fixation in the photic zone and governed by excess phosphorus and Fe availability as
previously used in Canfield (2006) and Boyle et al. (2013). In addition, we simulated primary production that
is dependent on OMZ-associated $N_2$ fixation, which in contrast to the classical $N_2$ fixation scenario is
independent of a Redfield-based nitrogen deficit with $N_2$ fixation being active as long as phosphorus and Fe
are available in concentration > 0 (Bombar et al., 2016;Löscher et al., 2014). One weakness of this model
simulation is that it includes Fe as potentially limiting nutrient for $N_2$ fixation, which is according to the
available datasets (Grand et al., 2015b;Chinni et al., 2019) not necessarily correct but may be valid as an
indicator for any other unrecognized micro-nutrient limitation. Consistent with the previous deep-time
models of Canfield (2006) and Boyle et al. (2013), our model exercise revealed that additional nitrogen supply
by $N_2$ fixation or other external nitrogen sources would generally exhaust the remaining traces of $O_2$ with
increasing upwelling (Fig. 6). According to our model, this would lead to denitrification, which is in line with
$O_2$-manipulated experiments as presented in Bristow et al. (2017) and consistent with the available isotope
records from the OMZ (Fig. S3). A weaker stratification (in the model depicted as increased upwelling fluxes)
would have the strongest effect on oxygen exhaustion and the onset of denitrification if primary production
is dependent on $N_2$ fixation in the photic zone, followed by OMZ-located $N_2$ fixation, and last by nitrogen
recycling. Given that OMZ regions are sites of massive nitrogen loss characterized by a nitrogen deficit in the
water column (Deutsch et al., 2007), the similar diazotroph community in the OMZ paired with an absence
of $N_2$ fixation in the euphotic zone suggest that OMZ-associated $N_2$ fixation is the most likely scenario. Thus,
nitrogen limited primary production in the BoB and in OMZs in general would be susceptible to changes in
stratification, with increased upwelling causing $O_2$ exhaustion. However, the fact that $N_2$ fixation is limited by
phosphorous supply via recycling in addition to upwelling and diffusive fluxes imposes an upper limit to $O_2$
depletion. Considering the potential $O_2$ sensitivity of OMZ diazotrophs based on the comparison with other



OMZs, the interplay between $O_2$ concentrations, stratification and $N_2$ fixation may act as a stabilizing
feedback on the BoB OMZ, preventing full $O_2$ depletion.
One factor possibly disturbing a possible stabilizing feedback is the external anthropogenic supply of nitrogen
to the northern Indian Ocean. This additional nitrogen source is projected to increase over the next decades
(Duce et al., 2008) potentially accelerating primary production in the future ocean including the BoB. An
atmospheric input in the range of 1.1 (model-based) to 1.6 Tg N yr-1 (observation based) has been reported,
which will likely increase in the future (Suntharalingam et al., 2019). This additional nitrogen fertilization
would cause the same effect as $N_2$ fixation in our model, thus exhausting the present traces of $O_2$ in the OMZ
rapidly. Until an increased supply of atmospheric or riverine nitrogen would become significant, changes in
water column stratification, however, likely impose the strongest control on $N_2$ fixation and primary
production, and thus on respiration, nitrogen loss processes and ultimately on the $O_2$ status of the OMZ in
the BoB.

**Conclusion**
We detected a diazotrophic community similar to those from other OMZ regions, however, we could not
obtain consistent evidence for active $N_2$ fixation in the BoB. Coming back to our original question 'No $N_2$
fixation in the BoB?' our data suggests 'No.'. In other OMZs, $N_2$ fixation has been observed to largely vary
temporally and spatially but never reaching rates comparable to oligotrophic open ocean systems such as
the Pacific gyres. Episodes of $N_2$ fixation, however, could be induced by changes in water mass dynamics,
riverine or atmospheric nutrient input. Resulting increased $N_2$ fixation and primary production would possibly
lead to $O_2$ exhaustion in the BoB, which otherwise doesn't become fully anoxic.
Previous observations describing the absence of nitrogen loss processes in the BoB were explained by the
remaining traces of $O_2$ (Bristow et al., 2017) and possibly by a nitrogen deficiency relative to carbon in the
organic matter pool. While we acknowledge that our dataset represents only a snapshot of the BoB's
biogeochemical setting, our observations may help to predict the future development of $N_2$ fixation in the
BoB and of the BoB OMZ with regard to increasing atmospheric dust deposition and ocean fertilization (Duce
et al., 2008), altered ocean circulation patterns (Yeh et al., 2009), and deoxygenation of the ocean as a
consequence of global warming (Schmidtko et al., 2017;Stramma et al., 2008).



**Code/Data availability:** Sequence data is available from Genbank, submission ID 2245434 and from NCBI's
sequence read archive, accession number SRR9696254. The model code and other biogeochemical data are
available from the Pangaea database (submission number PDI-21520 and 21522).
**Author contribution:** CRL carried analyzed the data together with WM, CRL ran the model simulations and
wrote the manuscript with substantial contributions from all co-authors.
**Competing interests:** The authors declare no competing interests.


**Acknowledgements**
We thank the captain and crew of the ORV *Sagar Kanya* for their support during sampling. We especially
thank the Ministry of Earth Sciences (MoES), India, for funding the research through the SIBER (INDIA) project
GAP2425 and for making RV *Sagar Kanya* available for this work. We thank J. Dekaezemacker and L. Piepgras
for sampling on board, providing nitrogen and carbon fixation rates and for helpful comments on the dataset,
and R. Boyle for providing the backbone model. We thank L. Bristow for helpful comments on an earlier
version of the manuscript, and we acknowledge E. Laursen for technical assistance, C. Callbeck and G. Lavik
for sampling, A. Treusch and M. Forth for providing access to subsamples for molecular analysis. We further
thank G. Krahmann for help with the analysis of fluorescence data from the CTD. This study was supported
by the H2020 program of the European Union (NITROX, grant #704272 to CRL) and the Max Planck Society.
Further funding was received from VILLUM FONDEN (Grant No. 16518; DEC) and the German Research
Foundation in the frameworks of the Cluster of Excellence 'The Future Ocean' and the Collaborative Research
Center SFB754.





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



**Tables:**

Table 1: Nitrogen fluxes in the BoB (Tg N yr$^{-1}$); N loss fluxes are given in black, N sources are given in in gray; DIN = dissolved inorganic nitrogen, PON = particulate organic nitrogen, IO = Indian Ocean. N$_2$ loss by denitrification was excluded by Bristow et al. (2017). Naqvi et al. (2010) reported possible N loss to the atmosphere in the form of N$_2$O.

| | Flux [Tg N yr$^{-1}$] | reference |
|---|---|---|
| Net exchange with the IO | 3.3 | Naqvi , 2010 |
| Sedimentary denitrification | 3 - 4.1 | Naqvi , 2008; Naqvi 2010 |
| PON burial | 1 | Naqvi , 2010 |
| Water column N loss to the atmosphere | 0 - 0.07 | Naqvi , 2010; Bristow, et al. 2017 |
| Atmospheric deposition | 0.5 - 1.6 | Naqvi , 2010; Singh et al., 2012; Suntharalingam et al., 2019 |
| Riverine/ land input | 0.4 - 4 | Naqvi , 2010; Singh et al., 2012; Krishna et al., 2016 |
| N$_2$ fixation | 0.6-11.3 | Naqvi , 2010; Srinivas & Sarin, 2013 |

**Figures:**

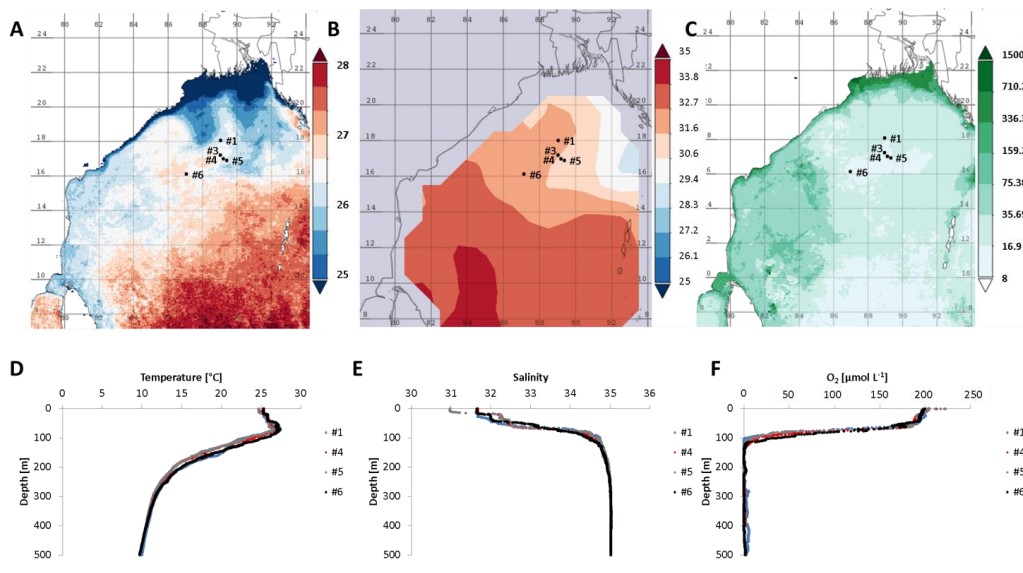

Fig. 1: Time averaged maps of (A) sea surface temperature (SST in °C (night only, 8-daily, 4 km resolution obtained from MODIS-Aqua), (B) sea surface salinity, (C) chlorophyll a concentration in 10$^{-2}$ mg m$^{-3}$, note the log scale (8-daily, 4 km resolution obtained from MODIS-Aqua). (D) CTD data-based water temperature in °C,



(E) salinity at the cruise stations. (F) O₂ (in µmol L⁻¹) over the top 500 m of the water column, data from
Bristow et al. (2017)

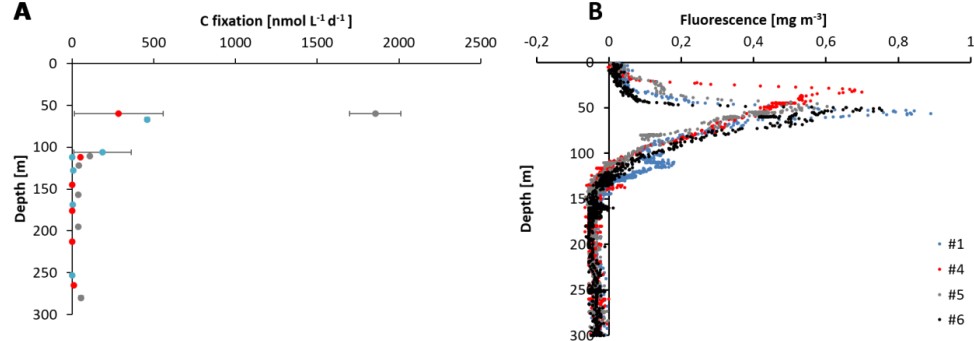

Fig. 2: (A) Carbon fixation rates at stations 1, 4 and 5, and (B) sensor-based fluorescence measurements from
station 1, 4, 5 and 6.

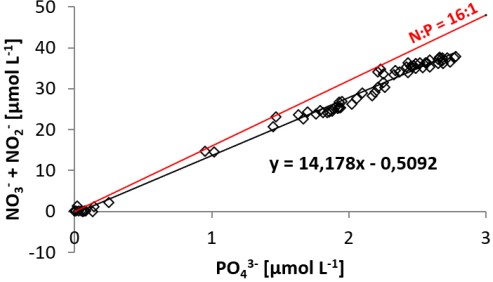

Fig. 3: N:P ratio at station 1, 4, 5 and 6, with the Redfield ratio of N:P = 16:1 indicated with a red line, the
negative intercept of the trendline indicates a deficit in dissolved inorganic nitrogen.

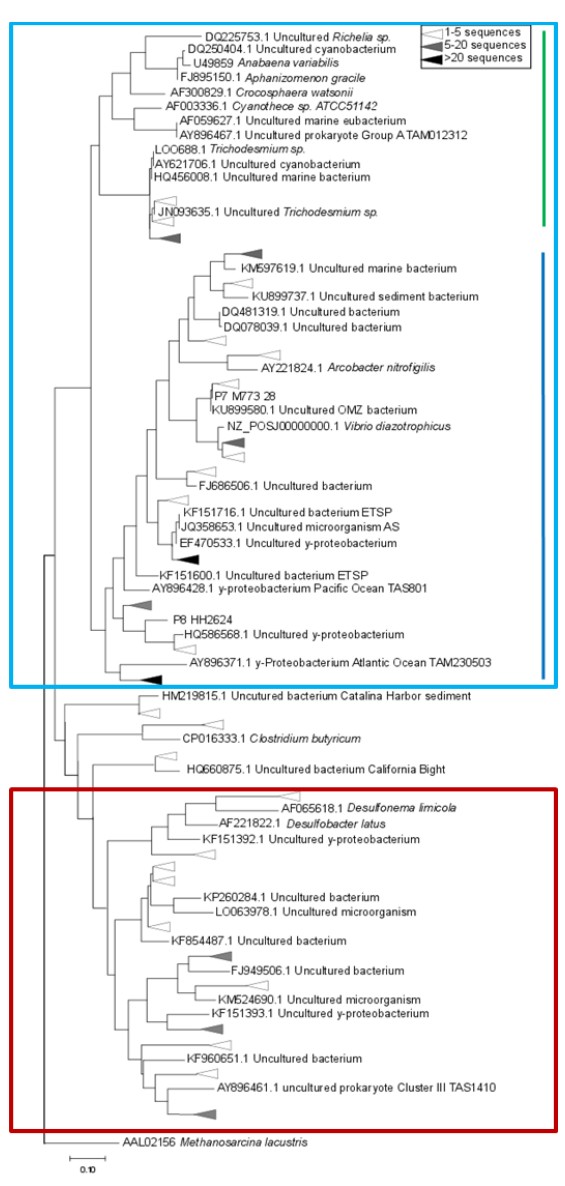


Fig. 4: Maximum likelihood tree of a 321 bp fragment of the *nifH* gene. Clusters identified by Sanger sequencing are indicated with triangles with colors denoting the sequence abundances in our dataset. The light blue box indicates Cluster I sequences including cyanobacteria (green line) and proteobacteria (blue line). Cluster III sequences are indicated with a red box.


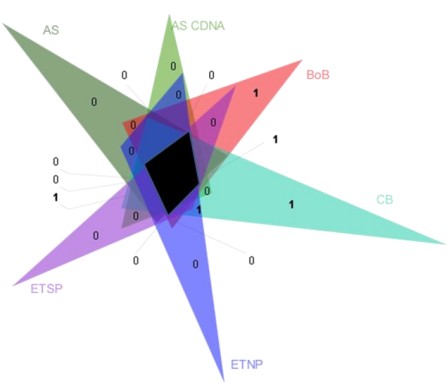


Fig. 5: Venn diagram of *nifH* clusters present in Arabian Sea DNA libraries (AS), and in cDNA libraries (AS
cDNA), clusters identified in the BoB, in $O_2$-depleted basins of the Californian Bay (CB), the eastern tropical
North Pacific (ETNP) and the eastern tropical South Pacific (ETSP). Clusters as depicted by triangles in Fig. S
6) were collapsed based on a 98% identity. The black area shows the clusters present in all OMZs. Numbers
indicate the individual clusters in fields which would otherwise appear unproportionally large.

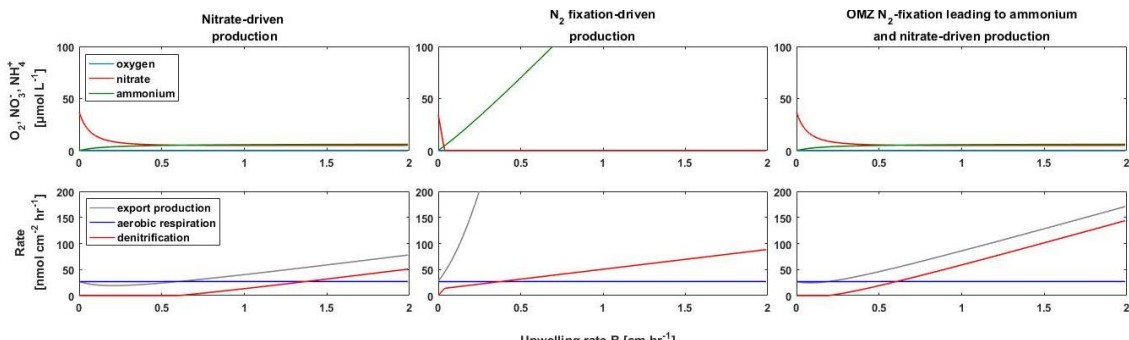

Fig. 6: Model of the response of the BoB OMZ to a weaker stratification corresponding to increased upwelling
in this model, under a non-$N_2$ fixation scenario with nitrate driven production, a photic zone $N_2$ fixation-
dependent primary production, and a scenario of $N_2$ fixation in the OMZ, which would result in ammonia
built-up and export to the productive surface if stratification becomes weaker.
