# Peer review of "No nitrogen fixation in the Bay of Bengal?"

_Biogeosciences, 2019_

## Referee Comment (RC1) · Anonymous Referee #1 · 6 Oct 2019

The manuscript by Löscher et al. addresses the important issue of assessing N2 fixation rates in the BoB OMZ. Few rate measurements and characterization of potential diazotrophic communities have been conducted in this region; therefore these observations are a valuable addition to the literature. I also appreciated the attempt to link controls on N2 fixation with the extent of oxygen depletion in the OMZ. I however I have a number of comments that should be considered prior to publication in Biogeosciences. These are listed below in the order I encountered them (not in order of importance).

Their hypothesis is that surface nutrient limitation restricts PP, the ensuing flux of sinking organic matter, and thereby oxygen removal at depth. However, the sampling focus is on N2 fixation at greater depths than the euphotic zone, which will accordingly not fuel surface PP without upwelling. This is generally prohibited in this strongly stratified regime. It is the level of N2 fixation in surface waters, not the OMZ, that will have the

greatest impact on PP and therefore (potentially) organic matter supply and respiration in the OMZ. As cited in the manuscript there are some observations indicating likely high surface nitrogen fixation by Trichodesmium in the Bay of Bengal (e.g. Sahu et al. 2017). I think this discrepancy between the hypothesis and the observations should be made explicit through the paper, including the abstract. However I do agree with the authors that the nitrate isotope data do seem to argue against this in the case of their observational time period.

The authors summarize some of the very high variability in N2 fixation in the Peruvian OMZ (i.e., detection limit to 840 umol N m-2 d-1); I think it could be useful to comment directly how this measured variability could provide relevant context to the BoB observations. i.e., how would this change the manuscripts conclusions? Indeed, that genomic signatures of diazotrophs were found suggests N2 fixation does occur in this system at some time points, which would be in line with sediment trap isotopic compositions.

I found a mixture of decimal points and commas (representing decimal points) in the text

Lines 179-180: Please provide some value ranges for SST and salinity

Please provide a description in the Methods section about the remote sensing images: where did these data come from (sensor, database), and what exact dates were used to produce the images in Fig. 1 (can also go in figure caption if preferred).

This also applies to the satellite-derived data in the SI, which includes phytoplankton types – it is important to know where this came from and the algorithms that were used to generate these.

Line 211: PP is a function of phytoplankton biomass and the biomass-normalized photosynthetic rate (depending on light availability, temperature etc), therefore lower POC biomass does not necessarily mean lower PP (as implied in the statement)

[Figure]

Line 217-218: But the N2 fixation measurements were not performed at the surface, they were performed at depth (>=60 m), where fixed nitrogen concentrations were presumably higher and therefore potentially removing this niche for N2 fixers.

Paragraph starting line 252 and Figure 5: Would it not be meaningful to include one or more non-OMZ sites in this analysis? i.e., to both indicate if (i) the OMZ sites have similarity between each other and (ii) are unique from non-OMZ sites?

Lines 324-327: I do not understand this sentence – if the model includes the potential for Fe limitation of N2 fixation, but the prescribed Fe concentrations are high, N2 fixation should not be effected by the Fe limitation term?

Model equations lines 8–10 of the SI: equations do not balance? E.g. I count 42 oxygens on the left hand side of the second equation and 51 oxygens on the right? Please check this and other equations!

Some more details on the model would be useful – i.e., does the model have any time iteration, is it ran until steady state for each upwelling value?

Figure 6: It is quite hard to see the different lines in the plot, e.g., I cannot see the blue oxygen line and whether it always stays at 0 for all upwelling values

Line 712: I don't understand the sentence: '. . .export to the productive surface if stratification becomes weaker'. Do you mean upwelling of ammonia to the surface, which then fuels more PP?

Lines 339-341: Regarding the sentence:

'However, the fact that N2 fixation is limited by phosphorous supply via recycling in addition to upwelling and diffusive fluxes imposes an upper limit to O2 depletion.'

But this cannot be the case in the actual BoB, as phosphate concentrations are in excess (not reported, but indicated by the negative intercept on the nitrate versus phosphate plot), yet there is still O2 present in the OMZ? Therefore the field data indicate P

availability in surface waters is not the cap on O2 depletion in the OMZ as suggested by the model?

It would be useful to briefly comment on how the physics of increased upwelling (or reduced stratification) might or might not increase ventilation of the OMZ? Are they completely decoupled?

―――――――――――――――――――――――

---

## Referee Comment (RC2) · Anonymous Referee #2 · 7 Oct 2019

Loescher et al present the first data set on Nitrogen fixation rates from the Bay of Bengal during winter. The Bay remains an enigma for biogeochemists as its western counterpart, the Arabian Sea, behaves very differently. Still, situated in the similar latitudes and divided by Indian Ocean, both basins have been hypothesized to be biogeochemically similar in the aspects of nitrogen inputs. However, until this study, no data existed on nitrogen fixation rates from the Bay, but a few studies have suggested unprecedented rates of nitrogen fixation in the Arabian Sea. Therefore, these data from the Bay are extremely valuable. Major conclusion of the study is that there was no nitrogen fixation in the oxygen minimum zones (OMZ) of the Bay, although the observed diazotrophic communities were similar to those reported to fix substantial nitrogen in other OMZ waters. This is intriguing. Overall, the manuscript reads well and extremely focused, but I have a few minor comments that authors might consider while revising the manuscript.

[Figure]

Since authors have focused mostly in the OMZ waters, it would be appropriate to change the title to "No nitrogen fixation in the oxygen minimum zones of the Bay of Bengal?" Most of the nitrogen fixation occurs in the surface waters in the Arabian Sea (and other regions), and it might still be there in the Bay. We do not know yet.

Line 66: There seems to be an extra closing parenthesis.

Line 77: 2,25 should be replaced by 2.25

Line 96: Cruise number should also be mentioned.

Line 202: It reads a bit awkward. How can Chlorophyll be compared to POC and stated as high or low? Both have different units.

Lines 215-217: "Based on the . . . . . . . . . . . . .cruise". These data are mostly from the deeper waters so nutrients are not limited for productivity. Moreover, concentrations of both nitrate and phosphate would be higher than required for any process. Not only that, nitrate is not needed for nitrogen fixation, so stating "BoB waters were nitrogen limited" is slightly misleading. I would suggest the authors to elaborate the significance of intercept, state limiting for which process, and also define the threshold limiting values for nutrients – for N2 fixation/C uptake, if required.

Line 248: It would be helpful to provide $\delta$15N range when denitrification or nitrogen fixation dominates.

Line 276 and 358: data are plural, so it should be "our data suggest"

Line 382: RV should be ORV

References: Kumar, S.P. and Prasanna Kumar, S. is one (same) author so such references should be clubbed.

Fig. 3: Equation on the should have decimal instead of commas. More appropriate will be to write y = 14.2 x − 0.6 as the data do not have precision of more than three significant figures.

Supplementary, Line 2 does not read well. Some grammatical mistake.

Fig. S3: 15N-NO3 should be replaced by $\delta$15N-NO3-, and delta 15N-PON should also be replaced by $\delta$15N-PON in the figure and caption both.

---

## Author Comment (AC1) · 6 Nov 2019

Dear reviewer, Thanks for this positive evaluation of our manuscript and for considering it a valuable contribution. I included your suggestions as follows and I believe they helped to make the manuscript clearer and better readable.

Line 66: There seems to be an extra closing parenthesis:

Author's response: This has been removed.

Line 77: 2,25 should be replaced by 2.25:

Author's response: This has been replaced.

Line 96: Cruise number should also be mentioned.:

Author's response: The cruise number is not available, but the exact dates were added

to better identify the cruise.

Line 202: It reads a bit awkward. How can Chlorophyll be compared to POC and stated as high or low? Both have different units.

Author's response: As for the ratio both, chlorophyll and POC was converted to mg m-3 following a study by Geider et al, 1997 in MEPS. However, the wording of the sentence was modified for better readability. It was indeed not very clear.

Lines 215-217: "Based on the . . .. . .. . .. . .. . .cruise". These data are mostly from the deeper waters, so nutrients are not limited for productivity. Moreover, concentrations of both nitrate and phosphate would be higher than required for any process. Not only that, nitrate is not needed for nitrogen fixation, so stating "BoB waters were nitrogen limited" is slightly misleading. I would suggest the authors to elaborate the significance of intercept, state limiting for which process, and also define the threshold limiting values for nutrients – for N2 fixation/C uptake, if required.

Author's response: This is a valuable point, and supports the view of another limiting nutrient such as or the limitation of processes by oxygen iron as discussed later on in the manuscript. For better readability, we changed the sentence to 'Based on the dissolved inorganic nitrogen (NO3- + NO2-) to phosphate (PO43-) ratio which has a negative intercept with the y-axis (Fig. 3; Benitez-Nelson (2000)), primary production in BoB waters appeared nitrogen limited during the cruise assuming Redfield stoichiometry.'

Line 248: It would be helpful to provide $\delta$15N range when denitrification or nitrogen fixation dominates.:

Author's response: The ranges have been added, however in an earlier place of the manuscript, where $\delta$15N signatures were presented first (l. 219): 'Consistent with this, $\delta$15N signatures of both the nitrate and the particulate organic nitrogen (PON) pool were only slightly decreased in the top 100 m of the water column to 5-8‰ (Fig. S3), thus not clearly supporting active N2 fixation which would be expected to create light

$\delta$15N signatures of -2- 2‰ (e.g. Dähnke and Thamdrup (2013)).'

Line 276 and 358: data are plural, so it should be "our data suggest":

Author's response: This is true, we changed this.

Line 382: RV should be ORV:

Author's response: This has been changed as suggested.

References: Kumar, S.P. and Prasanna Kumar, S. is one (same) author so such references should be clubbed.:

Author's response: Thanks for the hint, we changed this accordingly.

Fig. 3: Equation on the should have decimal instead of commas. More appropriate will be to write y = 14.2 x − 0.6 as the data do not have precision of more than three significant figures.:

Author's response: This has been changed as suggested.

Supplementary, Line 2 does not read well. Some grammatical mistake.:

Author's response: The sentence has been modified to 'The model framework is based on Canfield's 5-box model (Boyle et al. 2013; Canfield 2006), using available measurements for the BoB from our cruise and other literature (Tab. M1, the complete code will be released on Pangaea).'

Fig. S3: 15N-NO3 should be replaced by $\delta$15N-NO3-, and delta 15N-PON should also be replaced by $\delta$ 15N-PON in the figure and caption both.:

Author's response: Agreed and replaced

---

## Author Comment (AC2) · 6 Nov 2019

The cruise number has now been identified and added to the manuscript.

---

## Author Response (AR1)

Dear reviewer,

Thank you for considering our manuscript important and valuable for publication in Biogeosciences. We are grateful for your comments and suggestions, which we addressed in the revised version, we believe they largely helped to improve our manuscript.

Response to reviewer comments:

Their hypothesis is that surface nutrient limitation restricts PP, the ensuing flux of sinking organic matter, and thereby oxygen removal at depth. However, the sampling focus is on N2 fixation at greater depths than the euphotic zone, which will accordingly not fuel surface PP without upwelling. This is generally prohibited in this strongly stratified regime. It is the level of N2 fixation in surface waters, not the OMZ, that will have the greatest impact on PP and therefore (potentially) organic matter supply and respiration in the OMZ. As cited in the manuscript there are some observations indicating likely high surface nitrogen fixation by Trichodesmium in the Bay of Bengal (e.g. Sahu et al. 2017). I think this discrepancy between the hypothesis and the observations should be made explicit through the paper, including the abstract. However, I do agree with the authors that the nitrate isotope data do seem to argue against this in the case of their observational time period.

Response: This is a generally agreeable point, and the observation of Trichodesmium during other cruises is not contradictory but rather in line with our hypothesis of a water mass dynamic-dependent on-set of N2 fixation. For our cruise we were somewhat limited with regard to our coverage of the water column and thus we included isotope data which we read as an argument against N2 fixation during the time of the cruise in surface waters. The manuscript of Sahu et al is now more thoroughly discussed in our revised version of the manuscript.

The authors summarize some of the very high variability in N2 fixation in the Peruvian OMZ (i.e., detection limit to 840 umol N m-2 d-1); I think it could be useful to comment directly how this measured variability could provide relevant context to the BoB observations. i.e., how would this change the manuscripts conclusions? Indeed, that genomic signatures of diazotrophs were found suggests N2 fixation does occur in this system at some time points, which would be in line with sediment trap isotopic compositions.

Response: The observations from the OMZ off Peru are in so far interesting to compare to the BoB as they demonstrate first how $N_2$ fixation in OMZs can vary, but also they show a potential of a very similar community of $N_2$ fixers to actively fix $N_2$ (as opposed to them being largely inactive). If we would assume rates in the same range for the BoB, an additional N input would be provided which would, according to our model, be enough to cause full anoxia. This is a helpful point and has now been included into the revised version of the manuscript.

I found a mixture of decimal points and commas (representing decimal points) in the text

Response: This has been cleaned up.

Lines 179-180: Please provide some value ranges for SST and salinity

Response: Ranges have been added.

Please provide a description in the Methods section about the remote sensing images: where did these data come from (sensor, database), and what exact dates were used to produce the images in Fig. 1 (can also go in figure caption if preferred). This also applies to the satellite-derived data in the SI, which includes phytoplankton types – it is important to know where this came from and the algorithms that were used to generate these.

Response: The information has been added to the figure captions as suggested.

Line 211: PP is a function of phytoplankton biomass and the biomass-normalized photosynthetic rate (depending on light availability, temperature etc), therefore lower POC biomass does not necessarily mean lower PP (as implied in the statement)

Response: I have some difficulties with this comment because it reads tome as some disconnect between primary production and POC. But I overall agree that POC is not necessarily an indicator for primary production. This point has been presented right above from line 202 on. I revised the sentence in question to make it clearer, it now reads 'While our POC concentrations from DCM are one order of magnitude higher than the satellite-derived POC estimates (Fig. S2) from surface waters indicating that POC and primary production in surface waters was not higher than in the DCM, it must be noted that our measurements did not cover the entire mixed layer and are thus likely a rather conservative minimum estimate.'

Line 217-218: But the N2 fixation measurements were not performed at the surface, they were performed at depth (>=60 m), where fixed nitrogen concentrations were presumably higher and therefore potentially removing this niche for N2 fixers.

Response: This is true, and this is also the reason for the parallel presentation of nutrient data showing N depletion in the surface and of $\delta$15N data showing no $N_2$ fixation signal, both of which are thus supporting this claim.

Paragraph starting line 252 and Figure 5: Would it not be meaningful to include one or more non-OMZ sites in this analysis? i.e., to both indicate if (i) the OMZ sites have similarity between each other and (ii) are unique from non-OMZ sites?

Response: It would indeed be meaningful and relevant for a global assessment; however, the limit of the Venn diagram has been reached by including six datasets. The purpose of this plot was to show that the non-cyanobacterial clades in different OMZs are similar, non-OMZ areas would show cyanobacterial $N_2$ fixers in combination with heterotrophs, however, it has been pointed out previously that those heterotrophs re not active in non-OMZ areas (e.g. Turk-Kubo et al. (2014)).

Lines 324-327: I do not understand this sentence – if the model includes the potential for Fe limitation of N2 fixation, but the prescribed Fe concentrations are high, N2 fixation should not be effected by the Fe limitation term?

Response: This is correct, I intended to say that the model only includes Fe as limiting nutrient and doesn't have any other possibly limiting nutrients such as molybdenum of Vitamin B12 included. The sentence has been revised.

Model equations lines 8–10 of the SI: equations do not balance? E.g. I count 42 oxygens on the left hand side of the second equation and 51 oxygens on the right?
Please check this and other equations!

Response: Yes, a number got lost, here. Thanks for the hint.

Some more details on the model would be useful – i.e., does the model have any time iteration, is it ran until steady state for each upwelling value?

Response: The model does not have any time iteration, which is certainly a weakness given different elemental recycling times and is run into steady state for upwelling rates. The information has been added to the supplementary material.

Figure 6: It is quite hard to see the different lines in the plot, e.g., I cannot see the blue oxygen line and whether it always stays at 0 for all upwelling values

Response: We adjusted the lines in figure 6.

Line 712: I don't understand the sentence: '. . .export to the productive surface if stratification becomes weaker'. Do you mean upwelling of ammonia to the surface, which then fuels more PP?

Response: It meant to say that a stock of nitrogen would be build up, which could eventually be upwelled into surface waters and fuel primary production, there. The Figure caption has been rephrased for better readability.

Lines 339-341: Regarding the sentence: 'However, the fact that N2 fixation is limited by phosphorous supply via recycling in addition to upwelling and diffusive fluxes imposes an upper limit to O2 depletion.'
But this cannot be the case in the actual BoB, as phosphate concentrations are in excess (not reported, but indicated by the negative intercept on the nitrate versus phosphate plot), yet there is still O2 present in the OMZ? Therefore, the field data indicate P availability in surface waters is not the cap on O2 depletion in the OMZ as suggested by the model?

Response: This is actually very true; the statement made no sense in the context it was presented and has been removed.

It would be useful to briefly comment on how the physics of increased upwelling (or reduced stratification) might or might not increase ventilation of the OMZ? Are they completely decoupled?

Response: This would largely depend on the source of upwelled water. Our model distinguishes between deep water upwelling (ventilated) and intermediate OMZ water upwelling (not ventilated). We added a statement on the source of upwelled water to line 340.

Lines 215-217: "Based on the . . . .. . .. . .. . .cruise". These data are mostly from the deeper waters, so nutrients are not limited for productivity. Moreover, concentrations of both nitrate and phosphate would be higher than required for any process. Not only that, nitrate is not needed for nitrogen fixation, so stating "BoB waters were nitrogen limited" is slightly misleading. I would suggest the authors to elaborate the significance of intercept, state limiting for which process, and also define the threshold limiting values for nutrients – for N2 fixation/C uptake, if required.

Author's response: This is a valuable point, and supports the view of another limiting nutrient such as or the limitation of processes by oxygen iron as discussed later on in the manuscript. For better readability, we changed the sentence to 'Based on the dissolved inorganic nitrogen ($NO_3^-$ + $NO_2^-$) to phosphate ($PO_4^{3-}$) ratio which has a negative intercept with the y-axis (Fig. 3; Benitez-Nelson (2000)), primary production in BoB waters appeared nitrogen limited during the cruise assuming Redfield stoichiometry.'

Line 248: It would be helpful to provide δ15N range when denitrification or nitrogen fixation dominates.:

Author's response: The ranges have been added, however in an earlier place of the manuscript, where δ15N signatures were presented first (l. 219): 'Consistent with this, $\delta^{15}N$ signatures of both the nitrate and the particulate organic nitrogen (PON) pool were only slightly decreased in the top 100 m of the water column to 5-8‰ (Fig. S3), thus not clearly supporting active $N_2$ fixation which would be expected to create light $\delta^{15}N$ signatures of -2- 2‰ (e.g. Dähnke and Thamdrup (2013)).'

Line 276 and 358: data are plural, so it should be "our data suggest":

Author's response: This is true, we changed this.

Line 382: RV should be ORV:

Author's response: This has been changed as suggested.

References: Kumar, S.P. and Prasanna Kumar, S. is one (same) author so such references should be clubbed.:

Author's response: Thanks for the hint, we changed this accordingly.

Fig. 3: Equation on the should have decimal instead of commas. More appropriate will be to write y = 14.2 x – 0.6 as the data do not have precision of more than three significant figures.:

Author's response: This has been changed as suggested.

Supplementary, Line 2 does not read well. Some grammatical mistake.:

Author's response: The sentence has been modified to 'The model framework is based on Canfield's 5-box model (Boyle et al. 2013; Canfield 2006), using available measurements for the BoB from our cruise and other literature (Tab. M1, the complete code will be released on Pangaea).'

Fig. S3: 15N-NO3 should be replaced by δ15N-NO3-, and delta 15N-PON should also be replaced by δ 15N-PON in the figure and caption both.:

Author's response: Agreed and replaced.

[revised manuscript text omitted]
 based on Canfield's 5-box model (Boyle et al. 2013; Canfield 2006), using available measurements for the BoB from our cruise and other literature (Tab. M1, the complete code will be released on Pangaea). The model is classically based on the identification of the limiting nutrient for euphotic primary production under Redfield conditions. Primary production provides the basis for determination of export fluxes under consideration of respiration first using oxygen, followed by nitrate and sulfate according to the following stoichiometry:

$$C_6H_{12}O_6 + 6O_2 \rightarrow 6CO_2 + 6H_2O$$

$$2.5\ C_6H_{12}O_6 + 12NO_3^- \rightarrow 6N_2 + 15CO_2 + 12OH^- + 9H_2O$$

$$CH_3COO^- + SO_4^{2-} + H_2O \rightarrow H_2S + 2HCO_3^- + OH^-$$

[revised manuscript text omitted]

 **Figures**

[Figure]

Figure S1: Phytoplankton distribution in the BoB during the time of the cruise: (A) diatoms, (B) chlorophytes, (C) coccolithophores, and (D) cyanobacteria in mg m$^{-3}$. Data has been averaged from 15 Jan to 15 Feb, 2019, based on the MODIS-Aqua satellite product as available from https://giovanni.gsfc.nasa.gov.

[Figure]

Figure S2: Time-averaged (15 Jan to 15 Feb, 2019) POC distribution as monitored via MODIS-Aqua (https://giovanni.gsfc.nasa.gov) on an 8-daily basis, with a 4km resolution, POC concentrations in mg m$^{-3}$, concentrations in the cruise area were between 7.7 and 12.9 mg m$^{-3}$ and are consistent with our in-situ measurements.

[Figure]

Fig. S3: Both, (A) $\delta^{15}$N-NO$_3^-$ (data from Bristow et al., 2017) and (B) $\delta^{15}$N-PON show slightly lighter isotope signatures in the upper 100 m of the water column, however, this signal does not clearly indicate N$_2$ fixation.

---

## Author Response (AR3)

Dear editor, dear reviewers,

Thank you for the helpful suggestions and advice on our manuscript.

All the best

Carolin Löscher